# Assembly intermediates of orthoreovirus captured in the cell

Geoff Sutton [1,7], Dapeng Sun [2,7], Xiaofeng Fu[2,4,7], Abhay Kotecha [1,5,7], Corey W. Hecksel[3,6], Daniel K. Clare[3], Peijun Zhang [1,2,3✉], David I. Stuart [1,3✉] & Mark Boyce [1✉]

Traditionally, molecular assembly pathways for viruses are inferred from high resolution structures of purified stable intermediates, low resolution images of cell sections and genetic approaches. Here, we directly visualise an unsuspected 'single shelled' intermediate for a mammalian orthoreovirus in cryo-preserved infected cells, by cryo-electron tomography of cellular lamellae. Particle classification and averaging yields structures to 5.6 Å resolution, sufficient to identify secondary structural elements and produce an atomic model of the intermediate, comprising 120 copies each of protein λ1 and σ2. This λ1 shell is 'collapsed' compared to the mature virions, with molecules pushed inwards at the icosahedral fivefolds by ~100 Å, reminiscent of the first assembly intermediate of certain prokaryotic dsRNA viruses. This supports the supposition that these viruses share a common ancestor, and suggests mechanisms for the assembly of viruses of the *Reoviridae*. Such methodology holds promise for dissecting the replication cycle of many viruses.

[1] Division of Structural Biology, Wellcome Centre for Human Genetics, University of Oxford, Oxford OX3 7BN, UK. [2] Department of Structure Biology, University of Pittsburgh, Pittsburgh, PA 15260, USA. [3] Diamond Light Source Limited, Harwell Science and Innovation Campus, Didcot OX11 0DE, UK. [4] Present address: Department of Biological Science, Florida State University, Tallahassee, FL 32306, USA. [5] Present address: Thermo Fisher Scientific, Achtseweg Noorg 5, 5651 GG Eindhoven, The Netherlands. [6] Present address: Division of CryoEM and Bioimaging, SSRL, SLAC National Accelerator Laboratory, Stanford University, Menlo Park, CA 94025, USA. [7] These authors contributed equally: Geoff Sutton, Dapeng Sun, Xiaofeng Fu, Abhay Kotecha. ✉email: peijun@strubi.ox.ac.uk; dave@strubi.ox.ac.uk; mark.boyce@strubi.ox.ac.uk

Orthoreoviruses belong to a large family of viruses, the *Reoviridae*, which infect vertebrate, invertebrate and plant hosts and are responsible for economically and medically important diseases[1]. The mature virus particles are ~850 Å in diameter and are approximately spherical, with icosahedral symmetry. The particles harbour 10-separate genome segments and comprise two principle protein layers, the innermost composed of 120 copies of protein λ1 clamped together by 150 copies of σ2 (viral polymerases, λ3, are attached inside adjacent to the icosahedral fivefold axes). The outer layer consists of 600 copies each of μ1 and σ3. The array of μ1 and σ3 is interrupted at the fivefold axes by pentameric λ2 turrets which enzymatically cap the mRNA transcripts produced by the polymerase on egress from the particle. Finally, trimers of the fibrous cell adhesion protein σ1 emerge from the centres of the turrets. Unlike most RNA viruses these particles do not completely uncoat during cell entry, instead μ1, σ3 and σ1 are stripped off, maintaining in the cytoplasm a protein shell (known as the core) secluding the dsRNA genome from cellular pathogen recognition receptors. These particles are transcription-competent and, in the ribonucleoside triphosphate rich environment of the cytoplasm, they synthesise and extrude capped mRNAs, derived from the ten genome segments. These are translated to make the viral proteins which not only build new particles but also re-organise the volumes of the cytoplasm devoted to the manufacture of new virus (virus factories). These early stages of infection are relatively more understood than the later stages of assembly and egress[2]. Single-stranded gene segments are thought to interact with each other, facilitating the encapsidation of the complete viral genome[3], but how the multi-layered particles assemble, how the RNAs and replication enzymes are incorporated, what the signal for production of double stranded gene segments is, how assembly is completed by attachment of the final proteins and how particles leave the cell remain largely open questions.

Here we demonstrate that vitrification of infected cells followed by cryo-focussed ion beam (cryo-FIB) milling and cryo-electron tomography (cryo-ET) allows high resolution reconstruction of assembly intermediates of a mammalian reovirus, allowing atomic models to be constructed which throw light on some of these fundamental questions.

## Results

**In cell tomography of reovirus using FIB milling.** The first atomic structures for particles from the *Reoviridae* family were determined by X-ray crystallography[4–7], and advances in single particle cryo-electron microscopy (cryo-EM) have recently yielded structures at comparable or higher resolution for several members of the family[8–11]. Structural information is available for two types of purified orthoreovirus particles: the intact virus (we term similar particles that we see inside infected cells virion-like particles) and partially disassembled transcriptionally competent cores[7,12]. Cryo-EM has also allowed lower symmetry structures within the particle to be deconvoluted from the icosahedral structure of the protein shell, yielding insight into the spatial organisation of the polymerase and genome segments[9,13,14]. To link such atomic descriptions of stable purified particles to the sequential processes of virus assembly occurring inside infected cells, we grew MA104 cells on gold EM grids (Methods) and infected them with a mammalian orthoreovirus. Twelve-hours post infection the grids were plunged into liquid ethane and inspected by scanning electron microscopy (Methods and Fig. 1a). Cells were identified and regions chosen for cutting ~150–200 nm thick lamellae using cryo-FIB milling (Methods and Fig. 1b), a process that preserves high resolution information[15]. Milled lamellae were then analysed by cryo-ET and the tilt

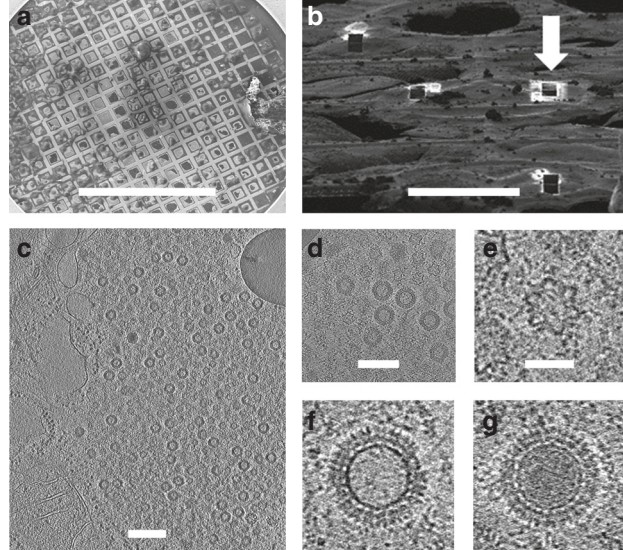

**Fig. 1 FIB milling and tomograms. a** SEM image of vitrified infected cells on EM grid prior to FIB milling. Scale bar 1 mm. **b** SEM image after FIB milling. White arrow shows one of the four-milled lamellae. Scale bar 100 μm. **c, d** Example tomogram of particles within cell (**d** shows a close up view). Scale bars 3000 and 1800 Å, respectively. **e–g** Representatives of stars, empty, and full virion-like particles, respectively taken from SIRT reconstruction by Tomo3D[44]. All three images are at the same scale and the scale bar corresponds to 450 Å. These images are representative of three lamella from which eight tomographic series were collected.

series analysed to yield three-dimensional reconstructions (Methods, Fig. 1c, d and Supplementary Movie 1). In total eight tomograms were collected, with particles from the five best used to generate the final structures. Overall, the attrition rate for the plunge–mill–collect workflow was high, with few lamellae proving useful. At 12-h post infection much of the cytoplasm contained viral factories crowded with virus particles enabling effective data collection even though the milling of lamellae was performed blind. A few enveloped particles were observed in some tomograms, consistent with co-infection with rotavirus, but the particles in virus factories analysed here are exclusively reovirus (see below). Unexpectedly, numerous star shaped particles were observed in the virus factories in addition to particles consistent with the expected virion architecture (Fig. 1e). In addition virion-like particles of two types, 'empty' and 'full', were observed (Fig. 1f, g). The 'full' particles presumably contain dsRNA.

**Sub-tomogram averaging of particles achieves high resolution.** Particles completely contained within the lamella were picked from five tomograms and classified into categories: 242 stars, 10 full and 65 empty virion-like particles (Methods). These classes were then subjected to sub-tomogram alignment and averaging, applying icosahedral symmetry, initially using PEET and subsequently emClarity[16] (Methods, Supplementary Table 1). Maps for the three categories are shown in Fig. 2a–c, Supplementary Fig. 1 and Supplementary Movies 2 and 3. The two virion-like particles are extremely similar, with the exception that the density level is much higher in the interior of the full particles (Fig. 2b, c and Supplementary Fig. 2). An accumulation of empty particles consistent with our observations has been reported for certain reovirus strains[17]. Higher resolution analysis, allowing for deviations from exact icosahedral symmetry, was then performed (Supplementary Table 1). For stars, analysis focused on the fivefold regions using emClarity[16] (Methods) yielded a reconstruction at 6.6 Å. For the empty virion-like particles,

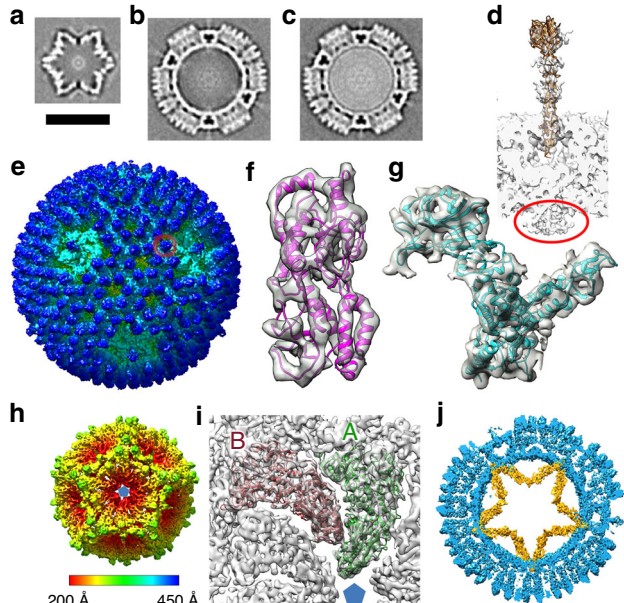

**Fig. 2 Sub-tomogram averaged reconstructions. a–c** Overviews of sub-tomogram averaging of stars, empty- and full virion-like particles respectively using IMOD/PEET[32]. Scale bar length 500 Å. **d** Cross-section through the fivefold of map (**b**). Circled in red is density attributed to the polymerase. The σ1 spike protrudes from the surface (a σ1 trimer is fitted). **e** Surface rendering of the emClarity map for the empty virion-like particle coloured by radius (scale bar as for panel h), a σ3/µ1 heterohexamer is circled in red. **f, g** Fits of the models (see "Methods") into the empty virion-like particle pseudo sixfold emClarity map, σ3 in purple, µ1 in cyan. **h**, **i** emClarity SLP map. **h** Coloured by radius. **i** Fit of λ1 subunits into the density, molecules A and B coloured green and red respectively. **j** Slice through the superposed sub-tomogram averaged emClarity maps; empty virion-like particle in blue and SLP in orange. The data were taken from five tomographic series (selected from 8 in total). Information on the number of particles used is given in Supplementary Table 1.

reconstructions focused on the fivefold regions and pseudo sixfold regions yielded reconstructions at 6.5 and 5.6 Å respectively (Methods, Supplementary Figs. 3 and 4a).

**Virion-like structures**. The virion-like structures can be directly compared to known structures of purified virions[7,12], which fit the density for the in situ structures extremely well (CCs for main chain 0.76 and 0.78 for the fivefold and pseudo sixfold regions of the empty particle, example fits to various maps are shown in Fig. 2d, f–g, Supplementary Fig. 4b, c and analysed in Supplementary Table 2). Thus purified virus particles provide a good model for the in-cell empty virion-like particle indicating that the internal RNA does not alter the fundamental organisation of the structure, thus at the fivefold axis on the exterior of the particle, there is evidence for the σ1 spike being attached (Fig. 2d). Although there is essentially no RNA in these empty particles there is evidence that the λ3 polymerase is attached at the expected position to the interior of λ1 close to the fivefold axis (Fig. 2d and Supplementary Fig. 1). However there is no evidence for the lattice-work organisation of the N-terminal 181 residues of λ1 B subunits on the inside of the particle which has been assumed to play a role in organising their assembly around the threefold axes[10], indeed there is no evidence for ordered structure for the first 259 residues of the B subunit (in the A subunit there is density for residue 241 onwards). It is possible that these termini play a role in interactions with, and possibly organising the genome in the mature virus.

**Star shaped particles comprising inner shell proteins**. The star shaped icosahedral particles comprise 120 major proteins (Fig. 2h, i) with additional density features on the outside. Since the inner shell of the virion-like particle is composed of 120 copies of λ1 we modelled these molecules into the density, initially fitting them as rigid bodies and then using Namdinator for molecular dynamics flexible fitting[18] (Methods). The quality of the fit was convincing, with principle secondary structural elements occupying strong density features (Fig. 2i) and the correlation between model (main chain) and experimental density increasing from 0.50 (single rigid body per molecule) to 0.79 (the quality of the agreement can be judged from Supplementary Fig. 4d and Supplementary Table 2). Unique features confirmed that these were reovirus particles (Supplementary Fig. 5). Broad features of the organisation of the molecules parallel the arrangement seen in the virus—five copies of λ1 (termed A molecules) cluster around each fivefold axis and five further copies of λ1 (termed B), orientated in a similar fashion to the A molecules, are located 35 Å further from the fivefold (Fig. 2i). Since the 60 copies each of the A and B molecules account for the large majority of the density for these particles we henceforth term them single-layered particles (SLPs). The maximum diameter of the SLPs is very similar to the corresponding layer of virion-like particles (Fig. 2j), however whilst the latter are roughly spherical the SLP is collapsed inwards at the icosahedral fivefolds by 100 Å, to give the characteristic star shaped cross-section (Supplementary Movie 4). Superposition of equivalent λ1 molecules between the virion-like particles and SLPs shows that the majority of the movement is caused by molecules A and B tipping inwards by ~35° (Fig. 3a, b). There are also significant conformational changes within the molecules, the largest being a hinge-like flexing of two parts of the subunit (the same hinge is used to adjust between the A and B conformations in the virus particles[4]). There is a 10° flex inwards of the fivefold adjacent part of molecule A and a similar 15° flex in molecule B (Fig. 3a, b). There are also significant conformational rearrangements throughout the A and B molecules (Supplementary Fig. 6) however, at the resolution of this analysis it would be premature to make a detailed interpretation.

**Conformational switching between the two particle types**. Due to the radically different configurations of the SLP and virion-like particle the contacts between the molecules differ markedly between the two particles. This is required to accommodate the proteins without compression towards the fivefold axes which are squeezed inwards in the SLP. The molecules rotate by some 35° and slide partially over each other like the blades of a camera iris. The two principle interfaces between the blades are between molecules, BA′ and AB, and there are a number of smaller interfaces, AB″, BB′, AA′ and AA″ (Fig. 3c). The largest interfaces (BA′ and AB) have contact areas of roughly 4400 and 4000 Å$^2$ compared to 5900 and 5500 Å$^2$ in the virion-like particle, respectively. Interface areas are given in Supplementary Table 3; due to the low resolution of the analysis the values for the SLP are very approximate (for methodology see "Methods"). Overall the contacts in the SLP are significantly weaker, with different residues involved. For the BA′ interface the iris blades are twisted such that at the tip there is a 30° rotation which allows the A′ molecule to dip inside the B molecule. In this region the interface is formed between what in the virion-like particle is part of the outer surface of A′ and the region of the B molecule which in that particle forms the inner surface. These contacts use completely different areas to those used in the expanded virion. For the AB interface the rotation of the iris blades again radically alters the interface towards the fivefold axes, in this case through an

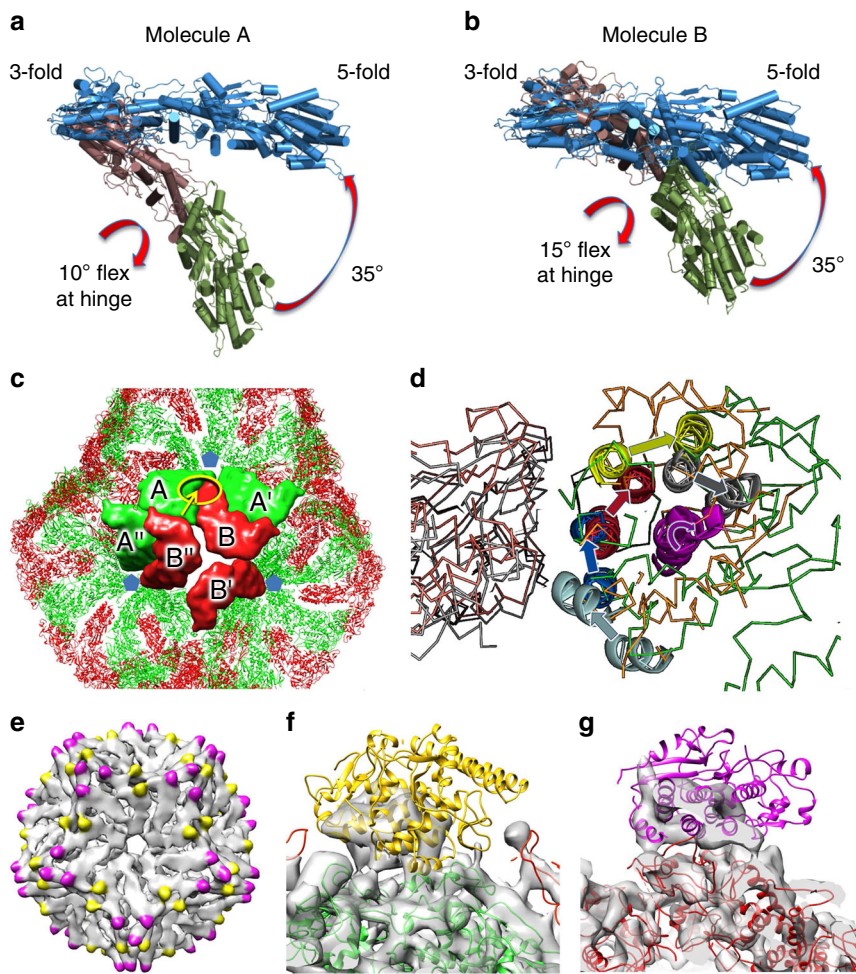

**Fig. 3 Rearrangement between SLP and virion-like particles. a**, **b** Side views of the conformational changes within the λ1 subunits between virion-like particle (blue) and the SLP where the two portions are coloured raspberry and forest, the latter portion being closest to the fivefold. **c** Interface areas between λ1 molecules in the SLP. Highlighted in yellow is the area shown in (**d**) with the arrow indicating the viewpoint. **d** Close up view of the ratchet of the AB interface. The A molecules are on the left. SLP molecules A and B coloured grey and orange, respectively. Molecules A and B for the virion-like particle are coloured pink and green, respectively. To show the ratchet more clearly the A molecules of each AB pair have been superposed. For the B molecules ratchetting helices and the pivot helix are coloured equivalently for the SLP and the virus (the pivot helix is shown in purple). Arrows show the rotation between the SLP and virus. **e** SLP density (PEET map) unaccounted for by λ1. Within the icosahedral asymmetric unit there are two distinct sets of density, one close to the twofold is coloured yellow, the other close to the threefold is coloured magenta. **f**, **g** close up of the densities shown in yellow and purple in (**e**). σ2 cartoons roughly positioned in the densities.

unusual ratchet mechanism. If the A molecule is considered fixed then the change corresponds to a 45° rotation around a pivot helix in B, with the effect that a bundle of helices surrounding the pivot helix rotate by a register of one helix (~12 Å), analogous to gears slipping by one tooth (Fig. 3d). In most cases interfaces are weakened in the SLP. However the >90° rotation leaves the AB″ and AA″ interfaces almost unchanged (2100–2200 and 1200–1000 Å$^2$ respectively). This is achieved by tucking the fivefold distal end of A under the edge of the B″ subunit. Overall the λ1 protein structure allows it to occupy two radically different configurations in the SLP/virion-like particle, but is likely not compatible with stable intermediate structures. As for the empty virion-like particles we see no trace of density for the first ~300 residues of either λ1 subunit, suggesting that if these residues do play a role in organising the assembly of the λ1 shell, as has been proposed[10], then it is via a loose association consistent with both the expanded and collapsed configurations that becomes ordered when the dsRNA packs against the interior of the core.

**σ2 only partially clamps the SLPs.** The 120 copies of λ1 are decorated with additional density at 120 sites (Fig. 3e). The density is poorly defined and only the region fairly close to the λ1 shell is seen, so it is not possible to unambiguously assign it to a particular protein however the overlap between the positions of these features and the attachment points of σ2 in the virus suggests that they are σ2[7]. The relative positions of σ2 in the SLP and virus are shown in Supplementary Fig. 7. In the virus there are 150 copies per particle, which attach at three distinct points within the icosahedral asymmetric unit. Two of these are in general positions, the other lies on the icosahedral twofolds. We name these three sites, twofold, threefold adjacent and A-hinge. In the SLP the threefold adjacent site remains occupied whilst the large rotation at the twofold separates the two half binding sites allowing two σ2 molecules to bind where only one can be accommodated in the expanded virion (Fig. 3f, g). In contrast, the A-hinge binding site is not occupied due to the iris like movement causing the adjacent B molecule to partially occlude the site. It is known that in the virion σ2 binding on top of the λ1 layer is

required for the stability of the expanded λ1 shell and it has been referred to as a clamp[7]. In the SLP it appears that σ2 only partially clamps the structure prior to expansion.

**SLPs contain do not contain RNA or visible polymerase.** There are no significant sites of unexplained density in the interior of the λ1 layer of the SLP that might be attributed to the λ3 polymerase. The λ3 polymerase could be disordered (the point of attachment to λ1 in the virion-like particle is a point of gross conformational change between the SLP and virion-like particle and so attachment would necessarily be different), or not found at every vertex, or be displaced away from the fivefold, which would lead to it being washed out due to fivefold averaging, or it could be simply not present at all. The interior volume of the SLP is 59% of that of the virion-like particle, thus the dsRNA genome segments cannot fit inside (the dsRNA in the core of the *Reoviridae* is already at a concentration of ~410 mg/ml[5]). In addition the density level in the tomogram is far lower than that in the full virion-like particles (Fig. 2a, c). It is believed that the genome is encapsidated in a single-stranded form. If single-stranded genome segments were present inside the SLPs then given the reduction in volume the density of RNA would still be 85% of that seen in the virus and the observed density levels do not support this (Fig. 2a, c).

## Discussion

The structures found here can be directly compared with assembly intermediates for other viruses. The *Cystoviridae* are a family of bacterial dsRNA viruses and although they are somewhat smaller and simpler than the *Reoviridae* (they possess only three genome segments) it has been proposed that they share a common ancestor[19]. The *Cystoviridae* have acted as a model system for understanding assembly and RNA packaging in dsRNA viruses[20–22] and the first stage in the assembly of these viruses is the formation of a highly indented SLP[23]. There is a striking similarity between the indented SLPs of phage Phi8 (representative of the *Cystoviridae*) and reovirus (Fig. 4). In both cases the structures are collapsed at the fivefold axes via hinge-like motions. Whilst for Phi8 the SLP clearly harbours copies of the viral polymerase, which is not seen in reovirus, in both cases the SLPs contain no genomic RNA, representing a pre-packaging state in assembly. The

*Cystoviridae* package ssRNA genome segments into preformed SLPs, in a well-defined order which is regulated by specific interactions of the genome segments with structural features on the outside of the particle which alter as the genome segments enter and inflate the particle[22,24]. The observation of an analogous pre-packaging SLP structure for reovirus suggests that incorporation of ssRNA might trigger expansion and addition of the outer protein components. The mechanism for ensuring the correct set of 10-genome segments is encapsidated for the *Reoviridae* is however very different from the *Cystoviridae*[20]. The far larger number of segments and the recent studies of genome segment selection in the *Reoviridae* indicate that the mechanism used to ensure encapsidation of a complete set of 10-genome segments includes RNA-RNA recognition between segments, possibly to form a concatamer of RNA which is then recognized for packaging[3,25–27]. If there is indeed some similarity in the assembly process, with collapsed particles being transformed into an expanded state by the recruitment of the RNA genome then the empty virion-like particles we observe would be dead-end products that have failed to package the genome. Indeed the proportion of empty virion-like particles observed is consistent with the known ~50% failure rate in packaging observed in strains which do not organize their virus factories in association with cellular microtubules[17]. The observation of only the SLP and completely assembled virion-like particles suggests that the conversion from the SLP to a virus-type structure occurs through the relatively rapid addition of all the remaining structural components of the virus. This is consistent with the description of an assembly intermediate that accumulates only when there is a temperature sensitive lesion in the λ2 turret protein delaying assembly events following the addition of λ2[28]—this expanded intermediate contained λ1, σ2 and also the λ3 polymerase protein, and lacked the complete dsRNA genome. The expanded shape and the additional presence of the λ3 polymerase protein indicate that this particle represents a step in the orthoreovirus assembly pathway between the abundant collapsed SLPs and completed virions observed in this study. Reovirus and rotavirus build their shells from a different set of proteins, and even those that are homologous are markedly different in sequence and structure. It seems unlikely therefore that a spatially separated contamination with rotavirus would have any effect on the reovirus structures we observe, however we cannot rule out that the co-infection perturbs cellular processes and hence the reovirus assembly pathway that we observe.

We believe that cryo-ET of suitably prepared infected cells will be a general vehicle for understanding virus replication-cycles in atomic and molecular detail, with our study already providing a reconstruction at 5.6 Å resolution from only 65 icosahedral particles. It is clear that further unexpected structures will be revealed which are simply not stable enough to be purified. Technical advances will optimise experimental procedures and improve resolution, and correlation with live-cell imaging and the use of mutant viruses will define the molecular context in the cell and illuminate the dynamics of the virus replication cycle.

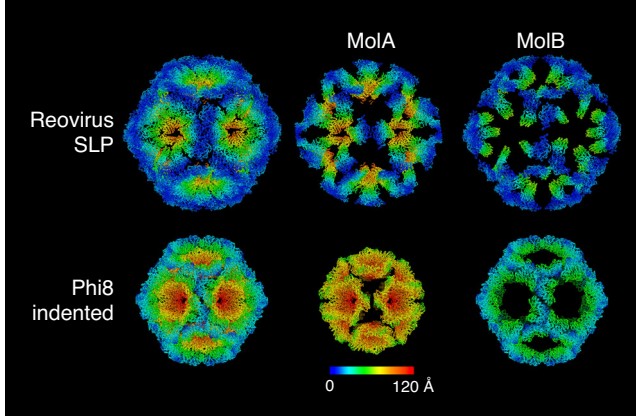

**Fig. 4 Similarity between the SLP and the indented form of Phi8 procapsid.** The structures are coloured according to the distance of equivalent Cα's between the indented and expanded forms of each virus. For clarity, only the indented state of each virus is shown. For both viruses molecule A moves more than molecule B, although it should be noted that there is greater movement between the two particle forms in Phi8.

## Methods

**Cell lines and virus**. MA104 cells were cultured in Dulbecco's modified Eagle's medium supplemented with 5% (v/v) foetal bovine serum at 37 °C in 5% $CO_2$. Mammalian orthoreovirus serotype 3 was used (sequence deposited for L1, S1 and S2).

**Cell preparation for cryo-EM**. Glow-discharged carbon-coated gold TEM grids (Quantifoil R2/1) were seeded with 33,000 MA104 cells and allowed to adhere for 4 h at 37 °C. Cells were infected at a multiplicity of infection of 10 and incubated at 37 °C. At 12-h post-infection the grids were washed three times in PBS, blotted

from the back side for 6–8 s and flash frozen in liquid ethane using a manual plunger.

**Focused ion beam milling**. The method was based on a previous report[15]. Grids were clipped into autogrid rims (Thermo Scientific) and loaded into a Scios DualBeam system (Thermo Scientific) using a Quorum cryo transfer station (PP3010T) and a custom built cryo stage cooled to −165 °C. Prior to milling, grids were coated with a ~2 μm layer of platinum using the gas injection system. Milling was performed in a step-wise fashion, using a 30-kV Ga ion beam, and beam currents of 300, 100, 49, and 30 pA to generate a lamella with a final thickness of ~150–200 nm. Milling progress was monitored with the scanning electron microscope using 3–5 kV beam and 25 pA beam current.

**Cryo-ET data acquisition**. Tilt series were collected on several positions from three different lamellae using a Thermo Scientific Krios electron microscope (eBIC Krios III) operating at 300 keV and equipped with a Gatan post-column energy filter (selecting a 20 eV window) on a K2 Summit direct electron detector (Gatan). Data acquisition was performed covering an angular range from −40° to +40° with 2° angular increments recorded automatically using the dose-symmetric tilting scheme[29] under low-dose conditions using SerialEM software[30]. Each tilt series was collected with a nominal defocus value between 4.3 and 5.9 μm. Each tilt was acquired as movies (containing 5 frames) in counting mode using a dose of 2 e$^-$/Å$^2$ per tilt. The total cumulative dose for each tilt series was 82 e$^-$/Å$^2$, with calibrated pixel size of 1.8 Å.

**Tomogram reconstruction and sub-tomogram averaging**. A total of eight tilt series were recorded from three lamellae, subjected to movie-frame alignment using Unblur[31], but without dose weighting since this is implemented in emClarity[16] at the final sub-tomogram averaging step. Initial analysis used IMOD/PEET[32,33], and these preliminary results are shown in Fig. 2a–c. For final analysis the frame-aligned tilt series were aligned and tomograms reconstructed in the framework of Appion[34] using Protomo[35]. The five tilt series of best quality (thinner lamella, minimum ice contamination) were selected and subjected to sub-tomogram alignment and averaging using emClarity. Tomograms were also reconstructed at binning 4 using SIRT for visualization. A total of 242 SLPs were picked from five tomograms and were subsequently aligned and averaged. The 12 fivefold vertices were identified in IMOD isosurface view from the averaged particle map allowing icosahedral symmetry to be applied to the whole SLP, resulting in an averaged map of SLP at 7.8 Å resolution. The fivefolds were then extracted from the symmetrized particle map and used as a reference for emClarity sub-tomogram particle picking by template matching. For emClarity processing, a total of 2683 SLP fivefolds were extracted. The 3D alignment procedures of SLP fivefolds were carried out gradually with binning of 4, 3, 2 and 1, each with 3 iterations. No manual mask was applied during the refinement. Dose-weighted filtering was applied at the final step. The FSC was calculated by the gold-standard method from even and odd data sets. The resolution of the SLP fivefolds is 6.6 Å at the 0.143 FSC cut off. The same process was carried out for a total of 65 empty virion-like particles, from those a total of 625 fivefolds and 3039 pseudo sixfolds were extracted, iteratively aligned and averaged. The resolution of these fivefold and pseudo sixfold reconstructions at the 0.143 FSC cut-off is 6.5 Å and 5.6 Å, respectively. A summary of cryoET data acquisition and data processing are presented in Supplementary Table 1.

**Model building, analysis and validation**. Models were derived from PDB 1EJ6, 2CSE, 1JMU and 3S6X. Initial fitting was performed by placing by hand into the density followed by Chimera[36] rigid body fitting. For the virion-like particle this was followed by molecular dynamics flexible fitting implemented in Namdinator[18,37]. For the SLP individual proteins (A and B) were fitted in Chimera and then each λ1 divided into two portions which were further rigid body fitted. Finally the fitting was optimised using Namdinator. Representative FSC curves for the fit of the models to the map are shown in Supplementary Fig. 4. Validation metrics (including correlation coefficients and FSCs) were determined using Phenix[38–40] and are presented in Supplementary Table 2.

Contact areas between protein subunits were calculated using the CCP4[41] program Areaimol[42,43]. Since contact surface calculations are very sensitive to coordinate error we enlarged the probe radius to 5.0 Å to reduce sensitivity and allow values to be compared with greater confidence.

**Reporting summary**. Further information on research design is available in the Nature Research Reporting Summary linked to this article.

## Data availability

Reovirus sequence data have been deposited at the NCBI, accession codes MT614295-7. Cryo-EM density maps of SLP fivefold and virion-like particle fivefold and pseudo sixfold have been deposited in EMDB under the accession codes EMD-22165, EMD-22166 and EMD-22164. The resulting atomic models have been deposited in the Protein Data Bank under the accession codes PDB-6ZTS, PDB-6ZTZ and PDB-6ZTY.

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

## Acknowledgements

We acknowledge Diamond for access and support of the Cryo-EM facilities at the UK national electron bio-imaging centre (eBIC), (proposal NR18477-18), funded by the Wellcome, MRC and BBSRC. The work was supported by the UK Medical Research Council (grant MR/N00065X/1 to D.I.S., M.B., A.K. and G.S.). A.K. was also supported by the Wellcome. Work was also supported by the National Institutes of Health (GM082251) and the UK Wellcome Investigator Award (206422/Z/17/Z) to P.Z., X.F. and D.S. Molecular graphics and analyses were performed with UCSF Chimera, developed by the Resource for Biocomputing, Visualization, and Informatics at the University of California, San Francisco, with support from NIH P41-GM103311. The work of the Wellcome Centre Human Genetics in Oxford is supported by a Wellcome core award 090532/Z/09/Z. This is a contribution from the UK Instruct-ERIC Centre.

## Author contributions

M.B. optimized cell culture and infection on grids, grew the virus and prepared virus infected cells. A.K. optimized grid preparation for FIB milling and prepared frozen grids. C.H. and A.K. carried out cryo-FIB milling, A.K. and D.C. collected tomograms and performed initial tomography reconstructions. X.F., D.S. and P.Z. performed tomography reconstruction, sub-tomogram alignment and averaging using IMOD/PEET, Appion/Protomo and emClarity. G.S. performed model fitting and analysis. D.I.S. conceived the experiments and with A.K. and M.B. designed the experiments. P.Z. with X.F. and D.S. designed and executed the workflows for cryo-tomography reconstruction and sub-tomogram averaging. D.I.S. and G.S. interpreted the structures and wrote the manuscript with input from all authors.

## Competing interests

The authors declare no competing interests.
