## [Peer Review File · Nature Communications]

Reviewers' Comments:

Reviewer #1:

Remarks to the Author:

This revised manuscript satisfies most of my previous concerns. However, like another reviewer, I remain highly concerned about the apparent "sloppiness" of having co-infected cells. The molecular tracings that suggest the structures better fit reovirus lambda-1 than rotavirus VP2 are slightly helpful. However, even if the structures truly are novel lambda/sigma structures, how do we know that the co-infection itself is not leading to these potentially novel structures?

Furthermore, the authors have included brief mention of conditionally-lethal phage mutants in their Abstract, but have still not put any of their potentially novel findings within any context with what is already known about orthoreovirus maturation and assembly from reovirus conditionally-lethal mutants.

Reviewer #3:

Remarks to the Author:

The authors provide an excellent account of a tour de force in defining the assembly intermediates of mammalian orthoreovirus serotype-3 (MRV-3) within infected cells. Reconstruction of the assembly intermediates was achieved via cryo-ET of particles within virus factories seen in lamellae derived from infected cells using cryo-FIB. These reconstructions reach subnanometer resolution with subtomographic averaging. Importantly, a proposed new single layer particle (SLP) was identified and characterized at the structural level, leading to a hypothesis that orthoreoviruses have a genome packaging mechanism into preformed procapsids similar to viruses of the Cystoviridae.

Technically speaking, it is an excellent piece of work. However, due to the uncertainty in the rotavirus contamination, it is rather unsettling about the ground truth on the biological conclusions in this report. Given the technical superiority of this work and the candid discussion of the admission of the contamination issue, I am supporting the acceptance of the paper provided that authors would add a cautionary note on possible scenario that star like particles might be formed because of the presence of the contaminating particles.

A few specific questions

What was the source and the strain of mammalian orthoreovirus serotype-3 used? Please cite in the materials and methods.

Line 189; what is a general position. Please clarify.

Line 199; why would the $\lambda 3$ polymerase be disordered? Rather, could an alternative be that the $\lambda 3$ polymerase is not found at every vertex and not be present in the subtomographic averaging?

Fig. 2j is somewhat confusing and a clearer explanation of what this figure represents is needed.

Extended Data Fig. 3; it would be helpful if the authors could provide a side-by-side figure that looks directly down the 5-fold and 6-fold axes of symmetry rather than us a lollipop to show the location of the 6-fold axis of symmetry.

Point by point response to Referees' comments:

Referee #1 (Remarks to the Author):

This revised manuscript satisfies most of my previous concerns. However, like another reviewer, I remain highly concerned about the apparent "sloppiness" of having co-infected cells. The molecular tracings that suggest the structures better fit reovirus lambda-1 than rotavirus VP2 are slightly helpful. However, even if the structures truly are novel lambda/sigma structures, how do we know that the co-infection itself is not leading to these potentially novel structures?

We apologize for the apparent "sloppiness" of having reovirus and rotavirus co-infected cells. But, we would like to convey the message to the reviewer that this actually strongly demonstrates the power of *in situ* tomography and subtomogram averaging. The untold story is that we initially considered that the subtomogram averages were rotavirus as we intended. However, none of the published rotavirus structures fit into the density maps at 5.6 Å - 6.6 Å resolutions where α -helices are clearly resolved. Even the size and mass of virions are different. In fact, we did not know the identity of the virus until we searched the whole PDB database for virus particles and found it matches the orthoreovirus genus. To confirm this we performed sequencing of several gene segments including serotype-determining gene sigma 1 which identified the virus as an isolate of mammalian orthoreovirus serotype 3. We think this is an excellent showcase for *de novo* identification of unknown subjects, albeit unintentionally.

Furthermore, the authors have included brief mention of conditionally-lethal phage mutants in their Abstract, but have still not put any of their potentially novel findings within any context with what is already known about orthoreovirus maturation and assembly from reovirus conditionally-lethal mutants.

We have inserted the following into the Results/Discussion section:

This is consistent with the description of an assembly intermediate that accumulates only when there is a temperature sensitive lesion in the $\lambda 2$ turret protein delaying assembly events following the addition of $\lambda 2$ (Hazelton and Coombs 1999, DOI: 10.1128/JVI.73.3.2298-2308.1999) – this expanded intermediate contained $\lambda 1$, $\sigma 2$ and also the $\lambda 3$ polymerase protein, and lacked the complete dsRNA genome. The expanded shape and the additional presence of the $\lambda 3$ polymerase protein indicate that this particle represents a step in the orthoreovirus assembly pathway between the abundant collapsed SLPs and completed virions observed in this study.

Referee #3 (Remarks to the Author):

The authors provide an excellent account of a tour de force in defining the assembly intermediates of mammalian orthoreovirus serotype-3 (MRV-3) within infected cells. Reconstruction of the assembly intermediates was achieved via cryo-ET of particles within virus factories seen in lamellae derived from infected cells using cryo-FIB. These reconstructions

reach subnanometer resolution with subtomographic averaging. Importantly, a proposed new single layer particle (SLP) was identified and characterized at the structural level, leading to a hypothesis that orthoreoviruses have a genome packaging mechanism into preformed procapsids similar to viruses of the Cystoviridae.

Technically speaking, it is an excellent piece of work. However, due to the uncertainty in the rotavirus contamination, it is rather unsettling about the ground truth on the biological conclusions in this report. Given the technical superiority of this work and the candid discussion of the admission of the contamination issue, I am supporting the acceptance of the paper provided that authors would add a cautionary note on possible scenario that star like particles might be formed because of the presence of the contaminating particles.

We appreciate the reviewer's positive comments. Regarding the uncertainty in the rotavirus contamination, please see the response to reviewer #1 above. We have expanded the discussion in the light of this.

We also believe that Extended Data Fig. 5. shows unambiguously that the SLP particle is composed from reovirus $\lambda 1$ and that the rotavirus equivalent protein does not contain sufficient residues to account for all the density.

A few specific questions

What was the source and the strain of mammalian orthoreovirus serotype-3 used? Please cite in the materials and methods.

The strain is unknown due to its unexpected discovery in a rotavirus stock. The sequencing allows us to state with certainty that it is mammalian orthoreovirus of serotype 3, and also showed that this isolate was not in the sequence database. It has now been deposited in Genbank and the database entries added.

Line 189; what is a general position. Please clarify.

We have added a definition to the text.

Line 199; why would the $\lambda 3$ polymerase be disordered? Rather, could an alternative be that the $\lambda 3$ polymerase is not found at every vertex and not be present in the subtomographic averaging?

We appreciate the review's comment. It is possible that $\lambda 3$ polymerase is not found at every vertex and not be present in the subtomographic averaging, or a single polymerase off-vertex being smeared out since we applied 5-fold symmetry to the map. We have included the discussion in the revised manuscript.

Fig. 2j is somewhat confusing and a clearer explanation of what this figure represents is needed.

We have clarified the figure legend.

Extended Data Fig. 3; it would be helpful if the authors could provide a side-by-side figure that looks directly down the 5-fold and 6-fold axes of symmetry rather than us a lollipop to show the location of the 6-fold axis of symmetry.

We appreciate the review's suggestion. We have revised Extended Data Fig. 3 to show figure panels looking directly down the 5-fold and pseudo 6-fold axes alongside.